# Neighborhood Physical and Social Environments and Social Inequalities in Health in Older Adolescents and Young Adults: A Scoping Review

**DOI:** 10.3390/ijerph20085474

**Published:** 2023-04-11

**Authors:** Martine Shareck, Eliana Aubé, Stephanie Sersli

**Affiliations:** Département des Sciences de la Santé Communautaire, Faculté de Médecine et des Sciences de la Santé, Université de Sherbrooke, 3001 12e Avenue Nord, Sherbrooke, QC J1H 5N4, Canada

**Keywords:** adolescent, built environment, emerging adult, neighborhood, physical environment, social environment, social inequality, young adult

## Abstract

Poor health and well-being are prevalent among young people. Neighborhoods may play a role in promoting good health. Little is known on if and how neighborhood characteristics affect health, and social inequalities therein, among young people. In this scoping review, we asked: (1) what features of the neighborhood physical and social environments have been studied in association with the physical and mental health and well-being of young people 15 to 30 years old; and (2) to what extent have social differentials in these associations been studied, and how? We identified peer-reviewed articles (2000 to 2023) through database and snowball searches. We summarized study characteristics, exposure(s), outcome(s) and main findings, with an eye on social inequalities in health. Out of the 69 articles reviewed, most were quantitative, cross-sectional, conducted among 18-year-olds and younger, and focused on the residential neighborhood. Neighborhood social capital and mental health were the most common exposure and outcome studied, respectively. Almost half of the studies examined social inequalities in health, mostly across sex/gender, socioeconomic status, and ethnicity. Evidence gaps remain, which include exploring settings other than residential neighborhoods, studying the older age stratum of young adulthood, and assessing a broader range of social inequalities. Addressing these gaps can support research and action on designing healthy and equitable neighborhoods for young people.

## 1. Introduction

Late adolescence and young adulthood, a period extending approximately between 15 and 30 years of age [1], is characterized by significant changes in health behaviors (e.g., diet, physical activity, tobacco, alcohol and other drug misuse) and health outcomes (e.g., physical and mental health, overall well-being) [2,3,4]. It is also a period of the life course that harbors distinct biological, cognitive, neurological and emotional developments [4], as well as social transitions such as entering higher education or full-time employment, moving out of the parental home, and establishing a family [5]. Health-wise, poor mental health affects approximately one in four young people worldwide [6], and 75% of mental health disorders are diagnosed before age 24 [3]. Physical activity also tends to decrease as youth transition from adolescence to young adulthood [7,8]. These trends suggest that increased preventive attention for physical and mental health should be prioritized for this age group [4,8].

Late adolescence and young adulthood are also marked by social inequalities in health, defined as systematic differences in health between groups that occupy unequal positions in the social hierarchy based on their wealth, power, race, or other dimensions of marginalization [9] such as socioeconomic status (SES), gender identity, ethno-cultural background, migrant status, disability, sexual orientation or other social determinants of health [4,10,11,12]. More specifically, social inequalities in health are differences whereby more disadvantaged social groups—such as groups with a lower SES, racial/ethnic minorities, women or gender minorities, and precarious migrants—systematically register poorer health outcomes than their more advantaged counterparts [9]. Social inequalities in health may, however, vary according to age group and to the social indicator and health outcome studied. For instance, in one study on the emergence of health inequalities in UK older adolescents and young adults, Sweeting et al. (2016) found there was relative equality in health between socioeconomic groups during adolescence, but that inequalities emerged at age 24 for physical health, and at age 30 for mental health, for both men and women. In this case, physical and mental health levels were poorest among young people not in education or work, and highest among those in non-manual occupations [13]. In the USA, Harris et al. (2006) found that inequalities by sex/gender and race/ethnicity widened from adolescence to young adulthood for a range of health outcomes. However, no single sex/gender or racial/ethnic group were consistently at an advantage or disadvantage; rather, social inequalities varied by health outcome [14]. Fortunately, this period of the life course represents a window of opportunity for intervention to significantly reduce future health and inequality burdens [15]. This is critical considering that inequalities in health behaviors and outcomes established before adulthood may track over time [16].

How cities are designed plays a role in improving health and reducing social inequalities in health [17,18]. Both the physical and social environments of neighborhoods are associated with health and social inequalities in health. Here, we define the physical environment as including natural and built features and resources such as urban design, walkability, greenspaces, public spaces and the retail environment, and the social environment as including area-level socioeconomic characteristics such as income or education levels as well as measures of collective functioning such as social capital and social cohesion. Several mechanisms have been postulated to explain the contribution that neighborhood environments may make to social inequalities in health. The unequal distribution of neighborhood features and resources across social groups may directly contribute to social inequalities in health. For example, lower SES groups may live in areas that are less walkable, have fewer greenspaces and lower social cohesion, which may negatively impact their health. Physical and social environmental characteristics may also differentially affect the health of different social groups: at similar levels of exposure to a given environmental feature, the health of more marginalized groups may be more impacted than that of their less marginalized counterparts [19].

Physical and social environment–health associations vary across the life course, which limits our ability to extrapolate associations from studies among children or older adults to adolescents and young adults [20]. The extent to which the neighborhood environment differentially affects older adolescents and young adults’ health based on their social characteristics also remains uncertain [21]. Our literature scan identified a recent scoping review on neighborhood–health associations among young adults 18–29 years old, but it largely focused on the conceptual and methodological underpinnings of this body of research without attending to the associations studied [21]. We also identified four systematic reviews on adolescents and young adults’ health and neighborhood greenspace [22,23], material deprivation and violence [24], and the public realm more largely (e.g., built form, greenspace and parks, services, neighborhood aesthetics, public spaces, and walkability/bikeability) [25]. Among these four reviews, only one included populations aged older than 20 years, which limits our knowledge of associations among young or “emerging” adults up to age 30 [26], and only two discussed potential health inequalities by sex/gender, ethno-cultural background or other social determinants of health [22,25]. Our existing knowledge is thus limited in its coverage of (1) the range of possible environment–health associations, (2) the entire spectrum of late adolescence and young adulthood, and (3) social inequalities in health. Painting a more comprehensive picture of neighborhood and health inequalities research among young people can help identify gaps in the literature, but also areas to focus on in equity-focused urban policy and practice.

## 2. Objectives

This scoping review aimed to address two questions: (1) what features of the neighborhood physical and social environments have been studied in association with physical and mental health and well-being among older adolescents and young adults 15 to 30 years old; and (2) to what extent have social inequalities in these associations been studied, and how?

## 3. Methods

### 3.1. Design

Given our objectives, we opted to perform a scoping review, a design adequate for reviewing a wide range of publication types pertaining to a broad topic [27]. We followed the PRISMA Scoping Review (PRISMA-ScR) guidelines as described by Tricco et al. (2018). These guidelines support a systematic approach to collecting and assessing the literature while allowing a wider range of publications to be considered for inclusion (i.e., inclusion criteria are not based on methodological quality as in systematic reviews). They also provide a clear step-by-step approach to conducting scoping reviews [28]. Our protocol is available online [29].

### 3.2. Data Collection

#### 3.2.1. Search Strategy

Variations of the search strategy were tested and validated by the main author (MS), a research assistant (JB) and a health sciences librarian, starting with an initial search of Scopus and Medline to map out the preliminary index terms and testing keywords, index/search terms, truncations, and wildcards as needed for the remaining identified databases. Following this iterative exploratory phase, we established a final list of search terms pertaining to the population, outcomes, exposures, and geographic contexts of interest (Appendix A).

We searched the following databases: Scopus, Medline, CINAHL, APA PsychInFo, Social Work Abstracts and SocINDEX with the Boolean operators “population search terms” AND “outcome search terms” AND “(physical environment exposure search terms OR social environment exposure search terms)” AND “geographic context search terms” (example provided in Appendix A). We restricted queries to abstracts except for Scopus queries that included both title and abstract. We conducted our last search on 9 February 2023. A second search phase involved snowball sampling from papers included in the full text screening. Potentially relevant references were identified and screened, first by title and abstract, then by full text.

#### 3.2.2. Selection Criteria

We searched for articles published between 1 January 2000 and 9 February 2023. We chose the year 2000 as the lower bound for the search strategy given the increase in peer-reviewed publications on neighborhoods and health around this time, which aligned with the 2003 publication of the book Neighborhoods and Health by Kawachi and Berkman [30]. Inclusion criteria were: original empirical research; quantitative, qualitative or mixed methods; written in English or French; including participants aged between 15 and 30 years. Exclusion criteria were: reviews, commentaries, conference abstracts and dissertations; and studies focused exclusively on 15-year-olds or younger or 30-year-olds and older. Outcomes and exposures of interest had to be mentioned in the title or abstract to be included in the full text review. We imported citations into Rayyan software, where duplicates were identified and removed. MS and two research assistants independently screened titles and abstracts; conflicts were resolved through discussion between reviewers. Full-text screening was carried out by MS and a research assistant.

#### 3.2.3. Data Extraction and Synthesis

Data were extracted by EA in Covidence software and independently checked by MS. Following our stated objectives, we extracted the following information from each publication using a custom data charting matrix: bibliographic (publication and journal titles, authors, year), study (design, location, methodological approach, time of measurement (if applicable), sample size), population (age, sex/gender, special characteristics), geography (setting within which exposures were measured (e.g., residential, school), area definition (e.g., census tract)), exposure definition and measurement, outcome definition and measurement, assessment of inequalities (yes/no and method), and main findings (main effects and subgroup effects, if applicable).

We first grouped results by methodological approach (quantitative, qualitative, mixed), then by setting. For each study, we identified outcomes and exposures, described associations, and when possible, summarized inequality assessment methods and results.

## 4. Results

### 4.1. Literature Search Results

Our combined searches identified 1112 references, which we reduced to 660 after removing duplicates. In the first round of screening (title and abstract), we identified 135 abstracts for full-text screening. After full-text screening, we excluded 66 publications because they did not meet eligibility criteria, resulting in a total of 69 articles included in this scoping review (see Figure 1 for flowchart) [20,31,32,33,34,35,36,37,38,39,40,41,42,43,44,45,46,47,48,49,50,51,52,53,54,55,56,57,58,59,60,61,62,63,64,65,66,67,68,69,70,71,72,73,74,75,76,77,78,79,80,81,82,83,84,85,86,87,88,89,90,91,92,93,94,95,96,97,98].

### 4.2. Study Characteristics

Study characteristics are summarized in Table 1. Of the 69 articles reviewed, 62 (89.9%) used a quantitative approach, and 59 (85.5%) had a cross-sectional design. The qualitative (*n* = 5) and mixed-methods (*n* = 2) studies used various methodological frameworks (e.g., participatory research) and a diversity of data collection methods such as focus groups, individual interviews, community mapping, photo mapping and photovoice. Most studies were conducted in North America (*n* = 31) (predominantly the United States (*n* = 27)) or Europe (*n* = 26). Sampling and recruitment were generally population-based (*n* = 35) or school-based (*n* = 30).

A wide range of age groups were studied, as shown in Figure 2. The majority of studies (*n* = 58; 84.1%) included participants between the ages of 14 and 18 years, and only 22 studies (31.9%) included participants aged older than 20 years.

### 4.3. Outcome and Exposure Definition and Measurement

Our search captured a broad diversity of health outcomes and environmental exposures. To aid description, we categorized outcomes and exposures based on the definition included within the article and/or their overarching construct, and classified them as “subjective” or “objective”. Subjective and objective measures of health outcomes and environmental exposure are worth considering separately, since they often assess different, yet overlapping, concepts, and they each have advantages and limitations. Subjective and objective environmental measures can also point toward distinct intervention and policy responses [99,100]. Subjective measures included those that were self-reported by participants or their parents (e.g., self-reported mental health or perceived neighborhood safety), while objective measures were based on diagnostic data or administrative and third-party data usually handled within geographic information systems (GIS) (e.g., greenspace coverage).

#### 4.3.1. Health Outcomes

Table 1 shows the subjective and objective health outcomes studied. Subjective or self-reported outcome measures were most common, with 63 of 69 studies (91.3%) relying on such measures, four studies (5.8%) using objective measures, and only two studies (2.9%) using both types of measures. Subjective outcome categories included “mental health”, “depression and depressive symptoms”, “well-being”, “emotional problems”, “behavioral problems” and “general health”. Broad categories such as “mental health” encompassed emotional and psychological distress, stress, and self-esteem, while “well-being” included general satisfaction with life, self-efficacy, and happiness. The most common categories of subjective health outcomes were “mental health” (*n* = 32), “well-being” (*n* = 18), “general health” (*n* = 15) and “depressive symptoms” (*n* = 17). As most studies investigated more than one outcome, the number of studies in each category summed up to 95, which was higher than the number of studies reviewed (*n* = 69). We categorized objectively measured outcomes into: “mental health” (e.g., psychiatric disorder diagnoses), “well-being” (e.g., emotional response (happy vs. sad)) and “general health” (e.g., morbidity, physiological measures such as body mass index (BMI), cortisol), with “general health” being the most studied (*n* = 3).

#### 4.3.2. Environmental Exposures

Of the 69 studies reviewed, 31 (44.9%) assessed subjective exposure measures, 24 (34.8%) used objective exposure measures, and 15 (21.7%) included both subjective and objective measures. We categorized subjective exposures in five groups: “neighborhood social capital” (e.g., neighborhood cohesiveness, sense of community) (*n* = 33), “natural and built environment” (e.g., distance and quality of greenspace, built environment quality) (*n* = 11), “neighborhood safety” (e.g., safety, problems) (*n* = 15), “neighborhood socioeconomic status” (e.g., perceived socioeconomic status) (*n* = 2), “neighborhood satisfaction” (*n* = 2), and “residential stability” (*n* = 1). Objective environmental exposures included “natural and built environment” (e.g., GIS-measured green and blue space) (*n* = 21), “neighborhood socioeconomic status” (e.g., average income) (*n* = 14), “residential stability” (*n* = 1), “immigrant density/ethnic heterogeneity” (*n* = 1), “inequitable housing practices” (*n* = 1), and “pollution” (*n* = 5). Table 1 details the full list of exposures associated with each category.

### 4.4. Settings Definition and Inequalities Assessment

The majority of studies (95.7%) focused on physical and social characteristics of the residential setting. Geographical areas were either self-defined (46.4%), defined with administrative data such as census tract, zip code or county boundaries (36.2%) or defined using buffers centered around study participants’ homes (17.4%). Social inequalities in health were examined in 31 of the 69 papers, with the majority assessing inequalities by sex/gender (*n* =23), followed by socioeconomic status (e.g., income, family affluence, education) (*n* = 8) and ethnicity (*n* = 6). Eleven studies assessed more than one type of inequality.

### 4.5. Associations Studied and Direction of Associations

We provide an overview of the environmental exposure–health outcome associations studied in Table 2, while Appendix A further details the direction of associations. The most common association, studied in 16 articles, was between “natural and built environment” exposures and “mental health” outcomes. Of these, 13 articles reported a significant positive association, one association was not statistically significant, one was not clearly stated, and one was explored in a qualitative study. “Natural and built environment” exposures were also often positively associated with “well-being” outcomes (*n* = 10). Of the 10 associations, three were either positive or negative but not statistically significant, and seven presented positive and statistically significant associations. “General health” outcomes were positively associated with “natural and built environment” in seven studies. Results were not clearly presented for one of these, and all the others were positively associated. Another frequently studied association was between “neighborhood social capital” and “depression and depressive symptoms” (*n* = 12). “Neighborhood social capital” was generally inversely associated with “depression and depressive symptoms”; however, four of these associations were not statistically significant. Other outcomes that were frequently positively associated with “neighborhood social capital” were “general health” (*n* = 9), “mental health” (*n* = 9) and “well-being” (*n* = 8).

### 4.6. Evidence on Social Inequalities in Health

Our second study objective set out to examine social inequalities in health. We found just under half of the 69 studies reviewed (*n* = 31) assessed whether associations between an environmental exposure and health outcome varied across different social groups (Table 3). Groups were defined by participants’ sex/gender (*n* = 23), socioeconomic status such as annual household income, family affluence or parental education (*n* = 8), ethnicity (*n* = 6), immigrant status (*n* = 3) and level of residential urbanicity (urban vs. rural) (*n* = 1). Different approaches, such as subgroup or stratified analysis (*n* = 18) and effect measure modification (*n* = 15), were used to assess social differentials in associations. The broad picture emerging from social inequalities in health assessment is that associations between environmental exposures and health outcomes may vary across sex/gender and ethnicity. For instance, 15 out of 23 studies assessing sex/gender inequalities reported significant differences in associations between groups, but no clear picture emerged with regard to specific exposure and outcome associations. Conclusions regarding inequalities across other social groups remained elusive, given that few studies have assessed them and that results were equivocal. These may also vary by exposure and outcome combinations. For example, only four out of eight studies found associations to vary by socioeconomic status, and these all related to greenspace and mental health or well-being.

## 5. Discussion

The primary aim of this review was to describe available evidence regarding the relationship between neighborhood physical and social environments and physical and mental health and well-being among older adolescents and young adults 15 to 30 years old. A secondary aim was to describe if and how social differentials in associations had been studied. We included 69 papers for review. The most commonly studied associations, in order of importance, were: “mental health” in association with “natural and built environment” exposure, “depression and depressive symptoms” in association with “neighborhood social capital”, and “general health” in association with “neighborhood social capital”. Most studies found significant positive associations between mental health and greenspace, and between general health and social capital.

Although we reviewed studies that did not exclusively include older adolescents and young adults (i.e., 65.6% of the studies also included participants aged 14 years and younger), the results provide a general idea of the association between environmental features and older adolescents’ and young adults’ health and well-being, and how it varies across social groups. In recent years, a few reviews have been published regarding neighborhood effects on adolescents’ and young adults’ health. Unlike our review—which included a broad spectrum of environmental exposures and physical and mental health outcomes—others have tended to focus on specific combinations of exposures and outcomes (e.g., greenspace and mental health). Five recent reviews focused exclusively on mental health and well-being [21,22,23,24,25], and some of our findings align with these studies. For instance, Fleckney and Bentley (2021) found that despite a wide range of greenspace definitions (leading to challenges in comparing studies), greenspace was significantly associated with mental health and well-being in adolescents, an association we also identified. In addition, like Fleckney and Bentley (2021), we found that blue space was not statistically significantly associated with older adolescents’ and young adults’ mental health and well-being. In their systematic review, Zhang et al. (2020) examined the association between well-being and greenspace among 10–19 year-olds and found a positive (though not always significant) trend. Their definition of well-being included emotional, mental, and psychological health, depression, anxiety, stress, mood, happiness, and pleasure [23]. On the other hand, Vanaken et al. (2018) found no significant association between greenspace and well-being among young adults [21,22,23,24,25]. Our findings of greenspace being associated with mental health are comparable to those of Zhang et al. (2020), even though we used a more fine-grained characterization of mental health and well-being outcomes [23].

To our knowledge, our review is one of the first to have studied both physical and mental health outcomes as well as inequalities therein. Our results suggest neighborhood exposures may have differential effects on health depending on individuals’ social characteristics, especially sex/gender and ethnicity. For instance, in Gutman et al.’s study (2004), neighborhood cohesiveness had a significant positive effect on depressive symptoms for women only. A study with young adult public housing residents found that more Latinx individuals reported being positively influenced by their community compared to other ethnic groups [70]. Neighborhood concerns may also vary between social groups. For example, one study found that neighborhood violence was more concerning to Latinx and Asian youth than Black and biracial participants [70]. Such differentials in concerns might potentially lead to health inequalities [101], but this was not tested per se. Differential associations by sex/gender and ethnicity have similarly been reported in other reviews, including that of Curtis et al. (2013), whose systematic review focused on neighborhood poverty, living conditions and social stressors and common mental disorders among 10–20 year-olds [24], and Fleckney and Bentley (2021), who found that girls may benefit less from proximate greenspace exposure than boys [25]. Other social determinants of health that could moderate the environment–health associations have been less studied. This highlights the importance of assessing neighborhood effects on health at the population level, but also across social groups within populations.

Our review also highlighted several evidence gaps. First we found relatively few studies that focused on young adults, especially those of 25 to 30 years of age, despite emerging evidence that the older adolescence and young adulthood period now extends into the late 20s [4,26,102]. This review was nevertheless informative regarding young adults’ health. In fact, one-third of studies reviewed did include young people aged 20 years or older. Results of individual studies tended to be similar independently of age group. For example, Mavoa et al. (2019) reported an inverse association between neighborhood greenness and depressive symptoms among 12- to 19-year-old participants [78], just as Dzhambov et al. (2018) noted that greenness was positively associated with mental health among 18 to 35 year-olds [64]. A few cross-sectional studies also provided results stratified by age group and found associations to be consistent across the entire age range of adolescence and young adulthood. For instance, higher greenspace presence was found to be associated with better mental health across the 11- to 20-year-old range in Bloemsma et al. (2022) [91], and for 15 to 25 year-olds in Dzhambov et al. (2018) [65]. On the other hand, Zhang et al. (2021) noted higher happy scores and lower sad scores among adolescents spending time in parks than among young adults [88].

In a related vein, longitudinal studies were rare, even though they could provide a better understanding of how the neighborhood environment influences health and health inequalities as adolescents transition toward young adulthood and further into adulthood [21,24]. Limited evidence from longitudinal studies reviewed here remains equivocal. For instance, Gutman et al. (2004) found that neighborhood social cohesion was significantly associated with depression in adolescence among women, but not men, an effect that did not extend into young adulthood [33]. Astell-Burt et al. (2014) found neighborhood greenspace to be inversely linked to psychiatric morbidity among women across the lifecourse, including in older adolescence and young adulthood, whereas in men, this inverse association was not seen for 15 to 20 year-olds but strengthened by age 30 [20]. Conversely, Engemann et al. (2019) reported a stronger association between neighborhood greenness and psychiatric disorder in adolescents 13 to 19 years old than in adults aged 20 years and older [72]. With so many developmental, social and neighborhood-related changes occurring during that critical lifecourse period, it would be valuable to follow older adolescents as they age to determine whether specific periods are more critical for environment–health associations, for the widening or narrowing of health inequalities, and for developing targeted health promotion interventions.

Second, more study is needed on the differential effect that the physical and social environments may have on health across social groups. This is important, given that neighborhoods are not necessarily equally experienced by different groups. Sex/gender, socioeconomic status and ethnic background were the main social characteristics studied, but it would be relevant to also explore social inequalities according to gender identity beyond man/woman differences and along the lines of sexual orientation, indigeneity, employment status or (dis)ability [103,104,105]. For instance, unemployed young people may spend more time in their local area and be more impacted by its features than fully employed individuals. Drawing on intersectional approaches would also be informative, since multiple identities such as being a woman in addition to being a recent immigrant or living on a low income may together lead to a different perception or experience of the urban environment and hence, to health [104,106].

Third, researchers should consider a wider range and combination of geographical settings, beyond the residential and school neighborhoods. Indeed, older adolescence and young adulthood are characterized by a diversification of social roles (e.g., leaving the parental home, entering higher education or full-time employment) [5] that coincide with decreased social and physical bonds to the residential neighborhood due to increased mobility, independence, and the development of relationships outside the residential area [107,108]. In the course of their daily activities, older adolescents and young adults may encounter environmental features and resources that can influence their health. These “activity space” exposures, which may also be socially patterned [109], should be explored in addition to the more common home or school neighborhood exposures [110]. Finally, the majority of studies reviewed here relied on subjective assessments of health, such as self-reported mental health and wellbeing. When used in combination with subjective exposure measures, results from these studies are prone to same-source bias, which makes causal relations hard to disentangle. More studies are thus needed that use objective measures of health outcomes.

Our scoping review had a number of strengths and limitations. Its main strength is that we mapped diverse outcomes and exposures into a broader conceptual grouping, which enabled us to synthesize the heterogenous evidence base on the relationship between environmental exposures and older adolescents’ and young adults’ health. While our categorization could be debated, disaggregated details for each publication reviewed are available as Appendix A (Appendix A), allowing for alternative syntheses by other researchers. Our aim to cover the literature broadly also had downsides. For instance, our findings regarding the effect of environmental exposures on physical health were inconclusive due to the lack of studies on physical health that emerged from our literature search. Despite the inclusion of the term “physical health” in our literature search strategy, only 17 references studied general health, and only two articles precisely aimed at measuring physical health. This small number of studies may be explained by the non-inclusion of specific physiological health terms, such as “obesity” or “body mass index (BMI)”, in our search strategy.

We also designed our literature search strategy to be broad enough to capture the age groups and populations we were interested in (i.e., older adolescents and young adults). However, as discussed above, different terms may be used in different fields to refer to this period of the lifecourse [26]. We may have overlooked publications that referred to young people in other ways or that did not name them per se [21]. We found relatively few studies focused on neighborhoods and health among older adolescents and young adults exclusively; rather, these were often included in samples of younger adolescents or general adults (e.g., 18 years old and over).

Finally, we did not review papers focused on health behaviors such as diet, substance use, or physical activity. Although these outcomes can be seen as being more proximal to neighborhood processes, lying along the pathway between neighborhood characteristics and the physical and mental health and well-being outcomes reviewed here, we limited the scope of the review to make it more manageable. A future step would be to conduct a similar review specifically focused on health behaviors among older adolescents and young adults.

## 6. Conclusions

Late adolescence and young adulthood is a key period for preventing illness and promoting health. This can be done through thoughtful design of neighborhoods and cities. However, given the dominance of studies examining mental health and well-being, our scoping review suggests that more research is warranted on associations between environmental exposures and physical health outcomes, as well as on the older age stratum of this period of the lifecourse. Furthermore, how neighborhood physical and social environments differentially affect the physical and mental health and well-being of older adolescents and young adults of different social groups is a notable knowledge gap that needs addressing. A comprehensive understanding of this matter is required for research to effectively guide policy and practice in designing healthy and equitable neighborhoods and cities for young people.

## 7. Future Directions

In light of this scoping review, future directions for research include examining if, and how, associations between the physical and social environments and physical and mental health vary across social groups other than gender/sex, ethnicity and immigrant status. Studying the older age stratum of young adulthood also merits attention.

## Figures and Tables

**Figure 1 ijerph-20-05474-f001:**
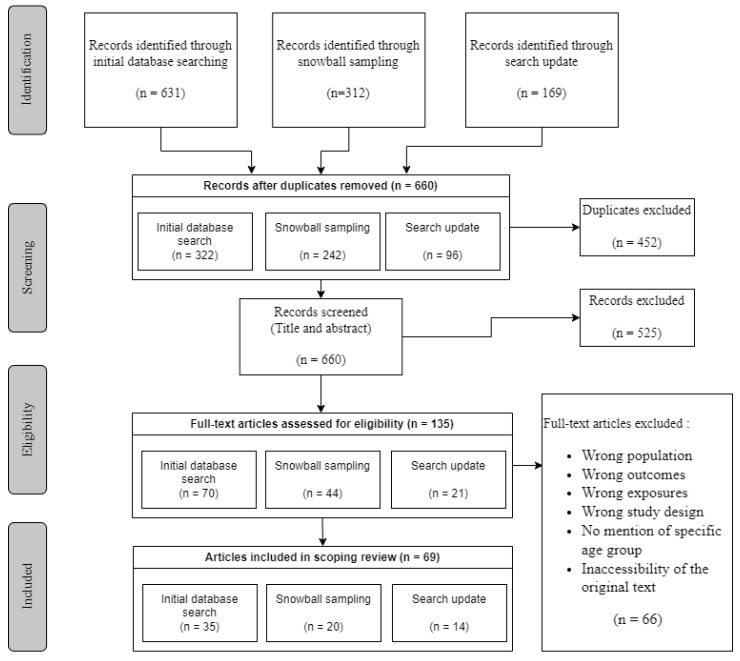
PRISMA flow chart.

**Figure 2 ijerph-20-05474-f002:**
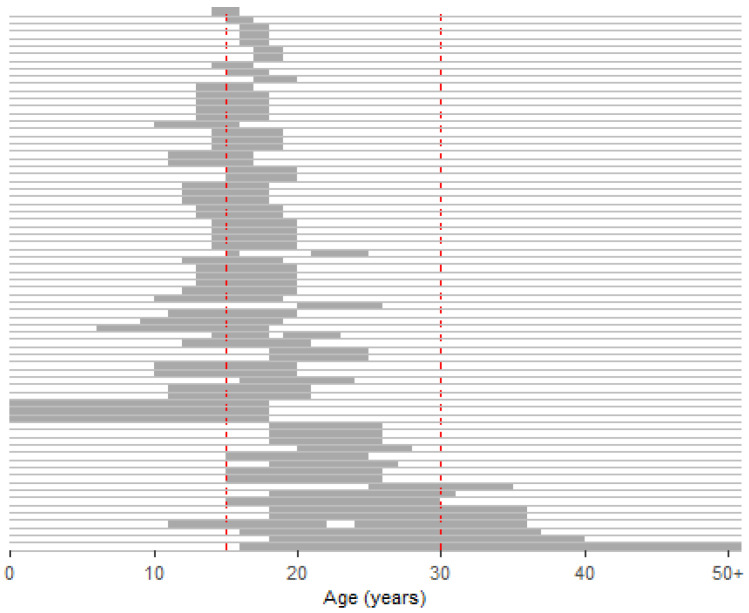
Age groups studied in the papers reviewed.

**Table 1 ijerph-20-05474-t001:** Descriptive characteristics of the 69 papers included in the scoping review.

			*n* (%)	
Approach	Quantitative		62 (89.9)	
	Qualitative		5 (7.2)	
	Mixed		2 (2.9)	
Study location ^a^	North America		31 (44.9)	
	Europe		26 (37.7)	
	Other		15 (21.7)	
Design	Cross-sectional		59 (85.5)	
	Longitudinal		10 (14.5)	
Recruitment	Population-based		35 (50.7)	
	School-based		30 (43.5)	
	Other		4 (5.8)	
Health outcomes	Subjective	Mental health (e.g., stress, psychological/emotional distress, self-esteem)	32 (46.4)	
		Depression and depressive symptoms (e.g., suicidal ideations and/or attempts)	17 (24.6)	
		Well-being (e.g., general satisfaction with life, self-efficacy, happiness, emotional response (happy vs. sad), flourishing)	18 (26.1)	
		Emotional problems	3 (4.3)	
		Behavioral problems (e.g., alcohol and drugs hazardous use)	4 (5.8)	
		General health	15 (21.7)	
	Objective	Mental health (e.g., psychiatric disorders)	2 (2.9)	
		Well-being (e.g., emotional response)	1 (1.4)	
		General health (e.g., morbidity, physiological measures: BMI, cortisol, etc.)	3 (4.3)	
Environmental exposures	Subjective	Neighborhood social capital (e.g., neighborhood cohesiveness, sense of community, social trust, collective socialization, collective efficacy, social cohesion, social support, social interaction, social control)	33 (47.8)	
		Natural and built environment (e.g., use, distance, quality, access, motivation to use, housing abandonment perception and built environment quality)	13 (18.8)	
		Neighborhood safety (e.g., violence, crime, safety, disorder, problems, danger, fear)	15 (21.7)	
		Neighborhood socioeconomic status (e.g., perceived neighborhood socioeconomic status).	2 (2.9)	
		Neighborhood satisfaction	2 (2.9)	
		Residential stability	1 (1.4)	
	Objective	Natural and built environment (e.g., greenspace, greenness, blue space, natural space, urbanicity, public natural space, intersection density, housing abandonment)	21 (30.4)	
		Neighborhood socioeconomic status (e.g., income inequality, economic deprivation, disadvantage, average income, poverty)	14 (20.3)	
		Residential stability	1 (1.4)	
		Immigrant density/ethnic heterogeneity	1 (1.4)	
		Inequitable housing practices	1 (1.4)	
		Pollution (e.g., air and noise pollution)	5 (7.2)	
Setting within which exposures were measured	Residential		66 (95.7)	
	School		2 (2.9)	
	Other		1 (1.4)	
Area definition	Self-defined		32 (46.4)	
	Administrative unit (e.g., census block or tract, zip code area)		25 (36.2)	
	Buffer (e.g., circular, road-network)		12 (17.4)	
Social inequality assessment	Yes ^b^	Sex/gender	23/31	31 (44.9)
		Socioeconomic status (income, education, affluence)	8/31	
		Ethnicity	6/31	
		Other (immigrant status, urbanicity)	4/31	
	No		38 (55.1)	

^a^ Three studies were conducted in more than one country. ^b^ Eleven studies explored more than one type of inequality.

**Table 2 ijerph-20-05474-t002:** Associations studied in the 69 articles included in the scoping review ^a^.

	Exposure	“Neighborhood Social Capital” ^b^	“Natural and Built Environment” ^c^	“Neighborhood Safety” ^d^	“Neighborhood Socioeconomic Status” ^e^	“Neighborhood Satisfaction”	“Residential Stability”	“Immigrant Density/Ethnic Heterogeneity”	“Pollution” ^f^
Outcome									
**“Mental health” ^g^**		Gutman 2004 [33], Glendinning 2007 [36], Novak 2015 [55], Cole 2019 [70], Kleszczewska 2019 [74], Lorenzo-Blanco 2019 [76], Hirota 2021 [90], Carrillo-Alvarez 2022 [92], Hunduma 2022 [93]	Astell-Burt 2014 [20], Glendinning 2007 [36], Larson 2022 [41], Dzhambov 2018 [64], Dzhambov 2018 [65], Roe 2017 [67], Cole 2019 [70], Colley 2019 [71], Engemann 2019 [72], Srugo 2019 [80], Wang 2019 [81], Franklin 2020 [82], Oswald 2021 [87], Shen 2020 [95], Stahlmann 2022 [96], Zewdie 2022 [98]	Gutman 2004 [33], Glendinning 2007 [36], Ivert 2013 [50]	Soobader 2000 [31], Chen 2006 [34], Ivert 2013 [50], Kleszczewska 2019 [74]	Cicognani 2008 [39]		Cole 2019 [70]	Dzhambov 2017 [63], Dzhambov 2018 [66], Franklin 2020 [82]
**“Depression and depressive symptoms” ^h^**		Wickrama 2003 [32], Day 2007 [35], Aslund 2010 [44], Wu 2010 [45], Delany-Brumsey 2014 [49], Lee 2015 [54], Pabayo 2016 [60], Estrada-Martinez 2019 [73], Kleszczewska 2019 [74], Lorenzo-Blanco 2019 [76], Oluwaseyi 2020 [83], Sadler 2022 [94]	Bezold 2018 [69], Mavoa 2019 [78], Zewdie 2022 [98]	Assari 2015 [53], Pabayo 2016 [60], Oluwaseyi 2020 [83], Kleszczewska 2019 [74], Mavoa 2019 [78]	Wickrama 2003 [32], Delany-Brumsey 2014 [49], Lee 2015 [54], Pabayo 2016 [60], Vilhjalmsdottir 2016 [62], Estrada-Martinez 2019 [73], Currier 2019 [86]	Estrada-Martinez 2019 [73]		Wickrama 2003 [32], Lee 2015 [54], Estrada-Martinez 2019 [73]	
**“Well-being” ^i^**		Day 2007 [35], Cicognani 2008 [39], DeClercq 2012 [46], Aminzadeh 2013 [47], Barnhart 2022 [68], Kleszczewska 2019 [74], Laurence 2019 [75], Malinowska-Cieslik 2019 [77], Benninger 2021 [85]	Huynh 2013 [48], Saw 2015 [58], Hogan 2016 [59], Teixeira 2016 [61], Roe 2017 [67], Mavoa 2019 [78], Zhang 2021 [88], Zhang 2022 [89], Bloemsma 2022 [91], Thompson 2022 [97]	Meltzer 2007 [37], Kleszczewska 2019 [74], Mavoa 2019 [78], Rigg 2019 [79], Benninger 2021 [85]	Day 2007 [35], Meltzer 2007 [37], Cicognani 2008 [39], DeClercq 2012 [46], Saw 2015 [58], Laurence 2019 [75]		Aminzadeh 2013 [47]		Bloemsma 2022 [91]
**“Emotional problems”**			Poulain 2020 [84]	Larson 2008 [40]	Soobader 2000 [31]				
**“Behavioral problems” ^j^**		Delany-Brumsey 2014 [49], Lorenzo-Blanco 2019 [76]			Soobader 2000 [31], Delany-Brumsey 2014 [49]		Delany-Brumsey 2014 [49]		
**“General health” ^k^**		Glendinning 2007 [36], Boyce 2008 [38], Borges 2010 [43], DeClercq 2012 [46], Marshall 2014 [51], Mmari 2014 [52], Novak 2015 [56], Novak 2016 [57], Benninger 2021 [85]	Glendinning 2007 [36], Maas 2009 [42], Mmari 2014 [52], Roe 2017 [67], Colley 2019 [71], Benninger 2021 [85], Thompson 2022 [97]	Larson 2008 [40], Benninger 2021 [85]	Soobader 2000 [31], Chen 2006 [34], Glendinning 2007 [36], DeClercq 2012 [46]				

^a^ Subjective and objective measures of exposures and outcomes are combined to facilitate synthesis. ^b^ E.g., neighborhood cohesiveness, sense of community, social trust, collective socialization, collective efficacy, social cohesion, social support, social interaction, social control. ^c^ E.g., greenspace, natural space, greenness, blue space, urbanicity, low intersection density, walkability. ^d^ E.g., violence, crime, safety, disorder, problems, danger, fear. ^e^ E.g., income inequality, economic deprivation, poverty, disadvantage, average income. ^f^ E.g., air and noise pollution. ^g^ E.g., stress, psychological/emotional distress, self-esteem, psychological health. ^h^ E.g., suicidal ideations and/or attempts. ^i^ E.g., general satisfaction with life, self-efficacy, happiness, emotional response (happy vs. sad), flourishing. ^j^ E.g., alcohol and drugs hazardous use. ^k^ E.g., morbidity, physiological measures (BMI, cortisol, health status).

**Table 3 ijerph-20-05474-t003:** Overview of studies assessing social differentials in neighborhood-health associations, by type of inequality studied ^a, b^.

Author (Year)	Outcome(s)/Exposure(s)	Inequality Assessment Method	Description of Finding
**Inequalities by sex/gender**			
Gutman (2004) [33]	Mental health/Neighborhood cohesiveness and problems	Subgroup analysis	Significant association between neighborhood cohesiveness and less depressive symptoms among girls but not boys.
Day (2007) [35]	Well-being and depression/Neighborhood social capital, social control, safety and SES	Effect measure modification	No significant interaction between neighborhood exposures and sex/gender.
Cicognani (2008) [39]	Well-being and stress/Sense of community, social support, area-level disadvantage	Subgroup analysis	Girls living in more disadvantaged town had higher well-being scores than boys.
Huynh (2013) [48]	Well-being/Natural space	Effect measure modification	No significant interaction between natural space and sex/gender.
Astell-Burt (2014) [20]	General health/Greenspace	Subgroup analysis	Greenspace associated with lower psychiatric morbidity among older adolescent and young adult women. Among men, greenspace not associated with psychiatric morbidity among 15–20 year-olds, but an inverse association strengthens by age 30.
Ivert (2014) [50]	Mental health/Neighborhood SES, collective efficacy and social disorder	Subgroup analysis	Poor collective efficacy associated with poorer mental health among boys but not girls.
Marshall (2014) [51]	General health/Neighborhood social capital	Subgroup analysis	Stronger associations between social capital and general health among girls compared to boys.
Mmari (2014) [52] (qualitative)	Health/Physical and social environments	Subgroup analysis	Feeling unsafe in the neighborhood mentioned as influencing health in Baltimore and Johannesburg (boys), and New Delhi, Shanghai and Ibadan (girls). Physical environment factors such as garbage, dirt, vacant housing and lack of recreation spaces mentioned by both boys and girls.
Assari (2015) [53]	Depression/Neighborhood fear of violence	Subgroup analysis	Increase in fear of neighborhood violence over a one-year period associated with an increase in depressive symptoms among men but not women.
Lee (2015) [54]	Depression and self-esteem/Neighborhood collective efficacy, Latino immigrant density and neighborhood poverty	Subgroup analysis	Neighborhood density of Latino immigrants associated with lower odds of depression among both male and female Latino immigrant youth, but not among non-immigrant Latino youth.
Novak (2015) [55]	Psychological distress/Neighborhood social capital	Effect measure modification	No significant interaction between social capital and sex/gender.
Novak (2015) [56]	General health/Neighborhood social capital	Subgroup analysis	No significant interaction between social capital and sex/gender.
Pabayo (2016) [60]	Depressive symptoms/Neighborhood disorder, danger, social cohesion, deprivation and income inequality	Effect measure modification	Girls living in more unequal neighborhoods had higher depressive symptoms than those living in more equal areas.
Dzhambov (2018) [64]	Mental health/Greenspace	Stratified analysis	Objective and perceived blue space measures associated with lower depressive symptoms among boys only.
Dzhambov (2018) [65]	Mental health/Greenspace	Effect measure modification and stratified analysis	No significant effect measure modification between greenspace and sex/gender.
Cole (2019) [70] (qualitative)	Resilience and mental health/Community member influences, building/land environment, diversity	Subgroup analysis	Boys reported more concerns about relations with police than girls, which might influence their health unequally.
Kleszczewska (2019) [74]	Depression, stress, satisfaction with life and self-efficacy/Neighborhood deprivation and social capital	Subgroup analysis	Social capital had the strongest protective effect for boys vs. girls in least privileged communities. Girls living in unsupportive neighborhood environments had very low satisfaction with life.
Malinowska-Cieslik (2019) [77]	Positive attitude/Neighborhood social capital	Subgroup analysis	No significant difference in associations between neighborhood social capital and positive attitude across sex/gender.
Poulain (2020) [84]	Emotional problems/Greenspace	Effect measure modification	No significant interaction between greenspace and sex/gender.
Zhang (2021) [88]	Emotional response/Greenspace	Effect measure modification	Male adolescents had higher happy scores than female adolescents.
Bloemsma (2022) [91]	Well-being/Greenspace and air and noise pollution	Effect measure modification	No significant interaction between greenspace and air or noise pollution and sex/gender.
Sadler (2022) [94]	Anxiety and depressive symptoms/Neighborhood social cohesion and inequitable housing practices (gentrifying, blockbusting, redlining)	Effect measure modification	Gentrification has a negative effect on social cohesion and well-being among girls but not boys. Blockbusted neighborhoods have lower social cohesion, leading to higher anxiety and depressive symptoms among boys but not girls.
Zewdie (2022) [98]	Psychological health (difficulties and depressive symptoms)/Greenspace	Effect measure modification	Greenspace inversely associated with total difficulties and depressive symptoms among boys but not girls.
**Inequalities by socioeconomic status (SES)**			
Aminzadeh (2013) [47]	Well-being and general mood/Neighborhood social capital	Effect measure modification	Membership in community organizations had stronger protective effect for students who were more, vs. less, socioeconomically disadvantaged.
Dzhambov (2018c) [65]	Mental health/Greenspace	Effect measure modification and stratified analysis	No significant interaction between greenspace and SES.
Srugo (2019) [80]	Psychological distress, mental health/Greenspace	Effect measure modification	No significant interaction between greenspace and individual SES or neighborhood deprivation.
Franklin (2020) [82]	Perceived stress/Pollution, greenness, and night light radiance	Effect measure modification	Association between artificial light at night and stress strongest among participants with lower vs. higher household income.
Poulain (2020) [84]	Emotional problems/Greenspace	Effect measure modification	No significant interaction between greenspace and SES.
Bloemsma (2022) [91]	Well-being/Greenspace and air and noise pollution	Effect measure modification	No significant interaction between greenspace and air or noise pollution and parental education.
Stahlmann (2022) [96]	Mental health/Built environment (social infrastructure places)	Stratified analyses	Stronger associations between social infrastructure and mental health among adolescents with high SES.
Zewdie (2022) [98]	Psychological health (difficulties and depressive symptoms)/Greenspace	Effect measure modification	Greenspace exposure associated with lower difficulties among those with an income vs. those without an income.
**Inequalities by ethnicity**			
Huynh (2013) [48]	Well-being/Natural space	Effect measure modification	No significant interaction between natural space and ethnicity.
Dzhambov (2018a) [64]	Mental health/Greenspace	Stratified analysis	Greenspace and blue space more strongly associated with mental health in non-Bulgarians than in Bulgarians.
Dzhambov (2018b) [66]	Mental health/Noise and air pollution	Subgroup analysis	Stronger inverse associations between noise and air pollution and mental health among non-Bulgarians than among Bulgarians.
Cole (2019) [70] (qualitative)	Resilience and mental health/Community member influences, building/land environment, diversity	Subgroup analysis	Asians more concerned about feeling unsafe in the community, while more Latinx youth emphasized feeling their community was like home and family. These differences in perceptions and concerns might unequally influence health.
Estrada-Martinez (2019) [73]	Depressive symptoms/Neighborhood satisfaction, SES and immigrant racial/ethnic composition	Effect measure modification and stratified analysis	Significant interaction between ethnicity and neighborhood satisfaction, racial/ethnic composition, and SES (e.g., low neighborhood satisfaction in adolescence associated with increases in depressive symptoms into adulthood for Mexicans and Puerto Ricans, and with lower levels of depressive symptoms in Cubans and other Latinos.
Shen (2022) [95]	Mental health/Built environment	Subgroup analysis	Built environment associations with mental health vary across ethnic groups (e.g., presence of libraries positively influences white youth’s mental health, presence of parks has a greater positive impact on Asian American youth’s mental health).
**Inequalities by immigrant status**			
Day (2007) [35]	Well-being and depression/Neighborhood social capital, social control, safety and SES	Effect measure modification	No significant interaction between neighborhood exposures and immigrant status.
Ivert (2014) [50]	Mental health/Neighborhood SES, collective efficacy and social disorder	Subgroup analysis	High perceived social disorder associated with mental health problems among Swedish background adolescents but not among those from immigrant backgrounds.
Lee (2015) [54]	Depression and self-esteem/Neighborhood collective efficacy, Latino immigrant density and neighborhood poverty	Subgroup analysis	Neighborhood density of Latino immigrants associated with lower odds of depression among Latino immigrant youth, both male and female, but not among non-immigrant Latino youth.
**Inequalities by level of urbanicity**			
Glendinning (2007) [36]	Mental health and general health/Social relations and trust and built environment	Subgroup analysis	Poorer perceptions of social and built environment characteristics associated with mental health in the city but not the smaller (rural) community.

^a^ The term sex/gender is used, as some studies used the term gender when actually measuring biological sex. ^b^ Studies may appear more than once in the table if they reported more than one type of inequality.

## Data Availability

Data are available upon request.

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
