# Peer review of "Neighborhood Physical and Social Environments and Social Inequalities in Health in Older Adolescents and Young Adults: A Scoping Review"

_ijerph, 2023, doi:10.3390/ijerph20085474_

Round 1

Reviewer 1 Report

Neighborhood built and social environments and health inequalities in older adolescents and young adults: a scoping review

While the abstract of the review provides a good overview of the study's aim, methodology, and main findings, there are a few areas where it could be improved:

Clearer research question: The abstract could benefit from a clearer research question that outlines the specific focus of the scoping review.

More specific details: The abstract could include more specific details about the study, such as the specific databases that were searched and the inclusion and exclusion criteria used to identify the 69 articles that were reviewed.

The abstract could be improved by providing clearer implications of the study's findings.

Introduction:

Provide a clearer research question: While the introduction outlines the general topic of the scoping review, it could benefit from a more specific research question.

Clarify the definition of social inequalities: The introduction briefly defines social inequalities in health but could provide a more detailed explanation. For instance, what are some specific dimensions of marginalization that may contribute to social inequalities in health, and how do these factors intersect with each other to create complex health disparities?

Provide more context on the importance of studying neighborhood environments: While the introduction mentions that neighborhoods can play a role in reducing health inequalities, it could provide more context on why studying neighborhood environments is important. For example, how do neighborhood characteristics interact with individual-level factors to impact health outcomes? What are some potential mechanisms by which neighborhoods influence health?

Summarize gaps in the literature: The introduction highlights some gaps in the existing literature, but could provide a more concise summary. For example, what are the key limitations of previous research on neighborhood environments and health among young adults, and how will this scoping review address these gaps?

Emphasize the relevance of the study: The introduction could benefit from emphasizing the relevance of the scoping review to a broader audience. For example, how will the findings of this review inform policy or practice related to neighborhood design and health promotion among young adults? What are some potential implications for reducing health disparities among marginalized groups?

Methods

Overall, the methods used in this scoping review appear to be sound and follow the PRISMA-ScR guidelines. However, there are a few areas where the authors could provide more information or clarification.

Firstly, it would be helpful if the authors provided more detail on the iterative exploratory phase of the search strategy. Specifically, it would be useful to know how many variations of the search strategy were tested and validated, and how the final search terms were chosen.

Secondly, the authors could provide more detail on the snowball sampling phase of the search strategy. For example, how many potentially relevant references were identified and screened, and how many of these were ultimately included in the review?

Thirdly, while the authors provide a comprehensive list of data extraction items, they do not provide any information on how data extraction was conducted. For example, were data extracted independently by multiple reviewers or by a single reviewer?

Lastly, the authors state that they grouped results by methodological approach (quantitative, qualitative, mixed), then by setting. However, it would be helpful if they provided more information on how they defined and operationalized "setting" in their analysis.

It is recommended that the protocol registration be made on the Open Science Framework (OSF) site, and that the link to the protocol be provided for transparency, rather than being available on request only.

Clearer research questions: While the review aims to chart the available evidence regarding the relationship between neighborhood built and social environments and physical and mental health and well-being among older adolescents and young adults, the research questions could be more specific and focused to guide the review process more effectively.

The presented discussion provides a comprehensive overview of the scoping review's findings. However, a few improvements can be made to enhance its clarity and specificity.

Provide more details about the methodology: It would be helpful to mention the specific search terms, databases searched, inclusion and exclusion criteria, and the screening process followed in selecting the studies for review.

Clarify the definition of "older adolescents and young adults": The age range of 15 to 30 years covers a broad developmental period, and it would be helpful to provide a more specific definition of the age group under study.

Provide a more detailed discussion of the findings: While the authors provide an overview of the most commonly studied associations, it would be useful to provide a more detailed discussion of the findings, including the strength of the associations, potential confounding factors, and the directionality of the associations.

Clarify the implications for policy and practice: The authors briefly mention the potential implications for policy and practice, but it would be helpful to provide more specific recommendations based on the findings of the review.

Reviewer 2 Report

Please, can you annex the PRISMA checklist and highlight the criteria taken into account in your study? 

Table 2 should be reformatted to make it easier to read. Perhaps you could put in foodnote or elsewhere the definition of examples such as "Neighborhood social capital" (e.g., neighborhood cohesiveness, sense of community, social trust, collective socialization, collective efficacy, social cohesion, social support, social interaction, social control)

You write Table 2. Associations studied in the 61 articles included in the review. I am not sure that the term "association" is the most appropriate one?

In the presentation of your results, tables 2 and 3 are given priority without any significant text supporting their contents. Describing your results in raw tables is restrictive.

What is the main information that the reader should retain in terms of results and in relation to your objectives?

Round 2

Reviewer 2 Report

Requested modifications have been correctly taken into consideration

Author Response

Thank you for the attention given to our manuscript.